

# What's for dinner? Diet and potential trophic impact of an invasive anuran *Hoplobatrachus tigerinus* on the Andaman archipelago

Nitya Prakash Mohanty[1,2] and John Measey[1]

[1] Centre for Invasion Biology, Department of Botany and Zoology, Stellenbosch University, Stellenbosch, South Africa
[2] Andaman & Nicobar Environment Team, Wandoor, Port Blair, Andaman and Nicobar Islands, India

## ABSTRACT

Amphibian invasions have considerable detrimental impacts on recipient ecosystems. However, reliable risk analysis of invasive amphibians still requires research on more non-native amphibian species. An invasive population of the Indian bullfrog, *Hoplobatrachus tigerinus*, is currently spreading on the Andaman archipelago and may have significant trophic impacts on native anurans through competition and predation. We carried out diet analyses of the invasive *H. tigerinus* and native anurans, across four habitat types and two seasons; we hypothesized that (i) small vertebrates constitute a majority of the *H. tigerinus* diet, particularly by volume and (ii) the diet of *H. tigerinus* significantly overlaps with the diet of native anurans, thereby, leading to potential competition. We assessed the diet of the invasive *H. tigerinus* ($n = 358$), and individuals of the genera *Limnonectes* ($n = 375$) and *Fejervarya* ($n = 65$) and found a significant dietary overlap of *H. tigerinus* with only *Limnonectes*. Small vertebrates, including several endemic species, constituted the majority of *H. tigerinus*, diet by volume, suggesting potential impact by predation. Prey consumption and electivity of the three anurans indicated a positive relationship between predator-prey body sizes. Individuals of *H. tigerinus* and *Fejervarya* chose evasive prey, suggesting that these two taxa are mostly ambush predators; individuals of Limnonectes chose a mixture of sedentary and evasive prey indicating that the species employs a combination of 'active search' and 'sit and wait' foraging strategies. All three species of anurans mostly consumed terrestrial prey. This intensive study on a genus of newly invasive amphibian contributes to knowledge of the impact of amphibian invasions, and elucidates the feeding ecology of *H. tigerinus*, and species of the genera *Limnonectes* and *Fejervarya*. We also stress the necessity to evaluate prey availability and volume in future studies for meaningful insights into diet of amphibians.

# INTRODUCTION

Accelerating rates of biological invasions (*Seebens et al., 2017*) and their consequent negative impacts (*Simberloff et al., 2013*) have led to increased efforts towards pre-invasion

Corresponding author
Nitya Prakash Mohanty,
nitya.mohanty@gmail.com

risk assessment and prioritization based on impact (*Blackburn et al., 2014*). Amphibian invasions have considerable detrimental impacts on recipient ecosystems (*Pitt, Vice & Pitzler, 2005*; *Kraus, 2015*), the magnitude of impact being comparable to that of invasive freshwater fish and birds (*Measey et al., 2016*). Impact mechanisms of amphibian invaders remain relatively understudied (*Crossland et al., 2008*) and are varied. Impact via predation and competition (*sensu Blackburn et al., 2014*) has been documented on invertebrates (*Greenlees et al., 2006*; *Choi & Beard, 2012*; *Shine, 2010*), fishes (*Lafferty & Page, 1997*), amphibians (*Kats & Ferrer, 2003*; *Wu et al., 2005*; *Measey et al., 2015*; *Liu et al., 2015*; but see *Greenlees et al., 2007*) and birds (*Boland, 2004*), though other taxa may also be affected (*Beard & Pitt, 2005*). Amphibian invaders may carry diseases (e.g., *Batrachochytrium dendrobatidis*; *Garner et al., 2006*; *Liu, Rohr & Li, 2013*) and cause reproductive interference (*D'Amore, Kirby & Hemingway, 2009*), apart from several other ecological impacts (see *Kraus, 2015*; *Measey et al., 2016* for detailed assessments).

However, reliable risk analysis of invasive amphibians still requires research on more non-native amphibian species, as the existing knowledge on impacts is mostly based on the cane toad *Rhinella marina* and the American bullfrog *Lithobates catesbeianus* (*Measey et al., 2016*). Comparisons of impact across taxonomic groups for management prioritization (*Blackburn et al., 2014*; *Kumschick et al., 2015*)  may also be impeded by the relatively understudied category of amphibian invasions as compared to other vertebrate invasions (*Pyšek et al., 2008*). This knowledge gap is further compounded by geographic biases in invasion research, with limited coverage in Asia and Africa (*Pyšek et al., 2008*); developing countries also have relatively less invasion research (*Nunez & Pauchard, 2010*; *Measey et al., 2016*).

An invasive population of the Indian bullfrog, *Hoplobatrachus tigerinus* (Daudin, 1802), is currently spreading on the Andaman archipelago, Bay of Bengal (*Mohanty & Measey, in press*). The bullfrog was most likely introduced in early 2000s and its exponential expansion has occurred since 2009, resulting in invasive populations on six out of the eight human inhabited islands of the Andaman archipelago (*Mohanty & Measey, in press*). 'Contaminants' of fish culture trade and intentional 'release' are likely to be the primary pathways of introduction and post-introduction dispersal, facilitating introductions from the Indian mainland and inter-island transfers (*Mohanty & Measey, in press*). The bullfrog has its native range on the Indian sub-continent encompassing low to moderate elevations in Nepal, Bhutan, Myanmar, Bangladesh, India, Pakistan, and Afghanistan (*Dutta, 1997*). The bullfrog has previously been introduced to Madagascar (*Glaw & Vences, 2007*), and possibly to the Maldives (*Dutta, 1997*) and Laccadive Islands (*Gardiner, 1906*). This large bodied frog (up to 160 mm) has high reproductive potential (up to 5,750 eggs per clutch; *Oliveira et al., 2017*) and is uncommon or absent in forested and coastal regions but occurs as a human commensal in plantations and agricultural fields (*Daniels, 2005*). It is considered a dietary generalist, feeding on invertebrates and even large anurans such as *Duttaphrynus melanostictus* (*Padhye et al., 2008*; *Datta & Khaledin, 2017*); however, quantitative diet assessment with adequate sample size across habitats and seasons is lacking (but see *Khatiwada et al., 2016* for diet of *H. tigerinus* in rice fields of Nepal).

*H. tigerinus* on the Andaman archipelago co-occurs with native anurans of the genera *Duttaphrynus*, *Fejervarya*, *Limnonectes*, and *Microhyla* (NPM unpublished data; *Harikrishnan, Vasudevan & Choudhury, 2010*). Given the large size of *H. tigerinus*, it is likely to feed on proportionately large prey, including amphibians and other vertebrates (*Datta & Khaledin, 2017*; *Measey et al., 2015*). The high volume of prey consumed by *H. tigerinus* (*Padhye et al., 2008*) may lead to direct competition with native anurans, especially under relatively high densities of *H. tigerinus* in human modified areas (*Daniels, 2005*). Although the diet of native anurans has not been assessed on the Andaman Islands, *Fejervarya limnocharis* is considered to be a generalist forager on terrestrial invertebrates (*Hirai & Matsui, 2001*); *Limnonectes* spp. are known to feed on vertebrates in addition to arthropods (*Emerson, Greene & Charnov, 1994*). This leads us to expect a high diet overlap of native frogs belonging to *Fejervarya* and *Limnonectes*, with the generalist *H. tigerinus*. In terms of size, *H. tigerinus* is much larger than native anurans of the Andaman archipelago (Fig. 1) and may impact the native anurans through both predation and competition.

Niche overlap, in combination with prey availability (electivity), can be used to assess trophic competition between species (e.g., *Vogt et al., 2017*). In addition to taxonomic evaluation and enumeration of the prey consumed, it is crucial to consider prey volume and frequency of prey occurrence to ascertain overall importance of a particular category of prey (*Hirschfeld & Rödel, 2011*; *Boelter et al., 2012*; *Choi & Beard, 2012*). Classification by functional type (hardness and motility of prey) is useful in understanding predator behaviour (*Toft, 1980*; *Vanhooydonck, Herrel & Van Damme, 2007*; *Carne & Measey, 2013*). Further, seasonality in prey availability may influence diet in amphibians (*Hodgkison & Hero, 2003*; *De Oliveira & Haddad, 2015*), therefore, there is also a need to assess diet across seasons, to fully capture the range of prey. Another important driver of prey choice may be the positive relationship between predator–prey body sizes (*Werner, Wellborn & McPeek, 1995*; *Wu et al., 2005*).

We aimed to assess the trophic impact of invasive *H. tigerinus* on the native anurans of the Andaman Islands through predation and potential competition. We carried out diet analyses of invasive *H. tigerinus* and native anurans, across four habitat types and two seasons, to ascertain the nature and magnitude of trophic impact. We hypothesized that (i) small vertebrates constitute a majority of the *H. tigerinus* diet, particularly, by volume and (ii) the diet of *H. tigerinus* significantly overlaps with the diet of native anurans, thereby, leading to potential competition. Additionally, we aimed to characterize the predation behaviour of these anurans in terms of electivity and predation strategy (ambush or active search).

## METHODS

We carried out the study in the Andaman archipelago for 6 months, from February to July 2017. The Andaman archipelago comprises nearly 300 islands (ca. 6,400$^2$) and is situated between 10°30′N to 13°40′N and 92°10′E to 93°10′E (Fig. 2). These islands are a part of the Indo-Burma biodiversity hotspot (*Myers et al., 2000*) with a 40% endemism level in herpetofauna (*Harikrishnan, Vasudevan & Choudhury, 2010*). The tropical archipelago
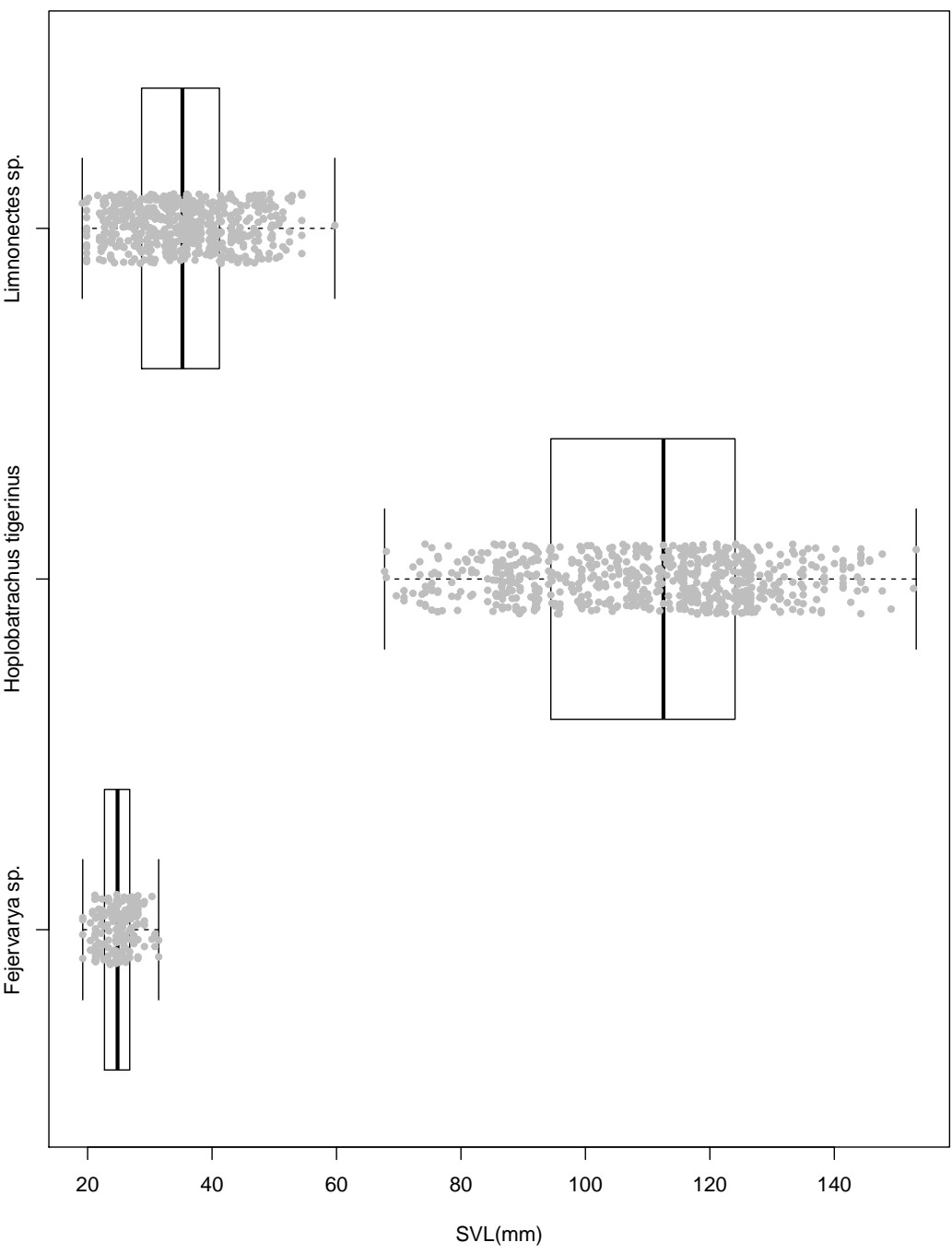

**Figure 1  Snout-vent length of three species of anurans used for diet assessment.** Individuals belong to the invasive Indian bullfrog *H. tigerinus*, the native *Limnonectes* spp. and *Fejervarya* spp., sampled at three locations on the Andaman archipelago.

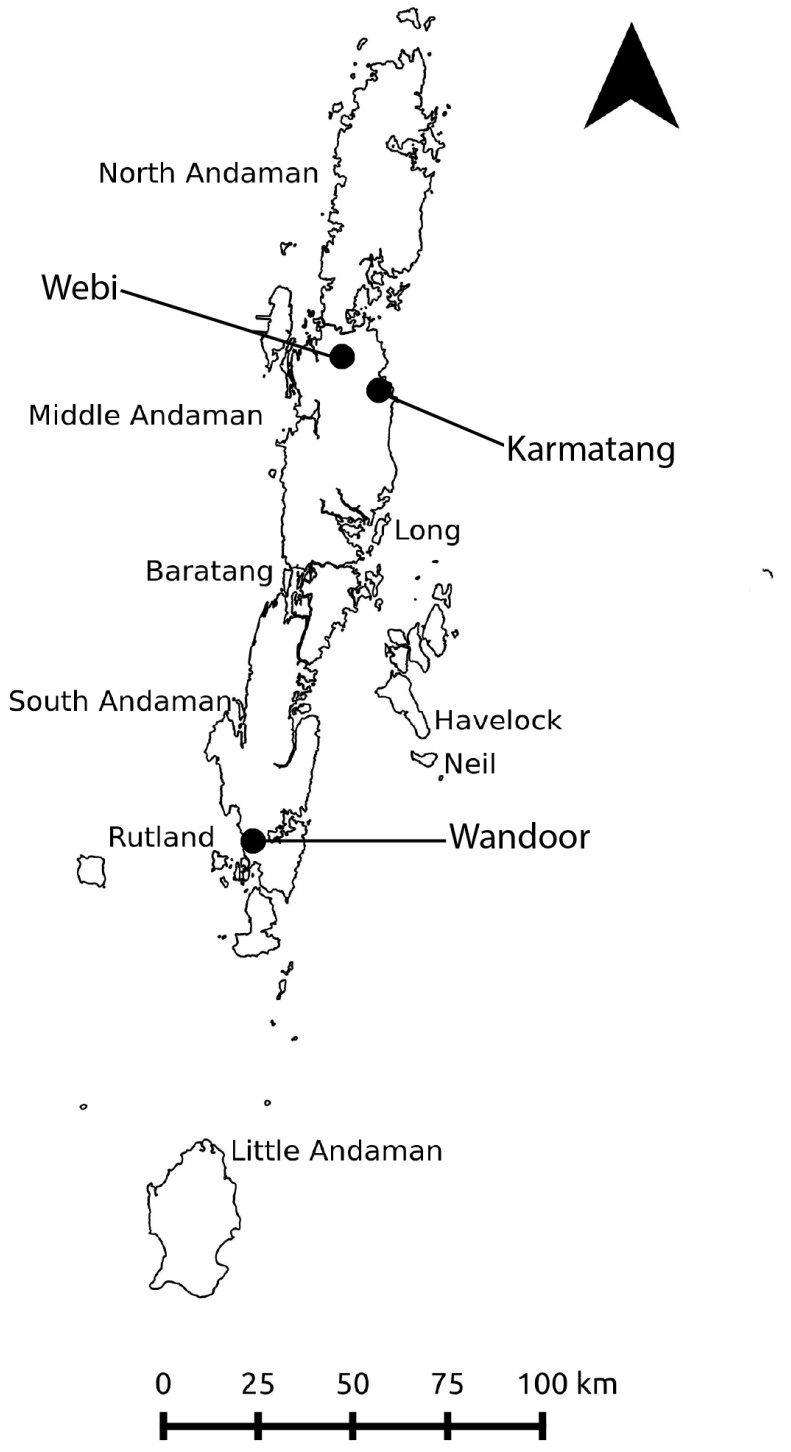

**Figure 2   Study area map showing the major islands of the Andaman archipelago and the three sampling locations.** Diet assessment of *H. tigerinus*, *Limnonectes* spp., and *Fejervarya* spp. were carried out from February 2017 to July 2017. Arrow indicates north.

receives an annual rainfall of 3,000 mm to 3,500 mm (*Andrews & Sankaran, 2002*); primary and secondary forests encompass nearly 87% of the entire archipelago (*Forest Statistics, 2013*), whereas the remaining human modified areas comprise of settlements, agricultural fields, and plantations. Of the nine species of native amphibians recorded, five species (*Ingerana charelsdarwinii*, *Blythophryne beryet*, *Microhyla chakrapani*, *Kaloula ghoshi* and *Fejervarya andamanensis*) are endemic to the Andaman Islands (*Das, 1999*; *Harikrishnan, Vasudevan & Choudhury, 2010*; *Chandramouli et al., 2016*; *Chandramouli & Prasad, 2018*), however, taxonomic uncertainties still persist (*Chandramouli et al., 2015*; *Harikrishnan & Vasudevan, 2018*). Post-metamorphic frogs of the range restricted *I. charlesdarwinii*, the semi-arboreal *B. beryet*, the arboreal *Kaloula ghosii* and the littoral *F. cancrivora* are unlikely to co-occur with *H. tigerinus* at present (*Das, 1999*; *Chandramouli, 2016*; *Chandramouli et al., 2016*). Thus, we constrained our choice for comparative species to those which were strictly syntopic. As the taxonomy of the Andaman amphibians remains in flux, we limited our identifications to the genus level for species belonging to the genera *Fejervarya* and *Limnonectes*, which are pending formal re-assessments (*Chandramouli et al., 2015*). Currently, *L. doriae*, *L. hascheanus*, *Fejervarya limnocharis*, *F. andamanensis*, and *F. cancrivora* are considered members of these two genera in the Andaman Islands (*Harikrishnan, Vasudevan & Choudhury, 2010*; *Harikrishnan & Vasudevan, 2018*). Hereafter, *Fejervarya* spp. and *Limnonectes* spp. are referred to as *Fejervarya* and *Limnonectes*, respectively.

We conducted the study at two sites (Webi and Karmatang) on Middle Andaman Island and at one site (Wandoor) on South Andaman Island (Fig. 2). We chose sites with moderately old invasions of *H. tigerinus* (more than 3 years since establishment; *Mohanty & Measey, in press*), assuming that a relatively longer time since establishment would indicate an adequate population to sample from. In each site, we established four 1 ha plots with varying land use-land cover types: agriculture, plantations (Areca nut and Banana), disturbed (logged) and undisturbed forest (minimal use). To capture the variation in diet with respect to seasons, we carried out the sampling in both dry (January to April) and wet (May to July) seasons, the latter coinciding with the south-westerly monsoon.

Our protocol was approved by the Research Ethics Committee: Animal Care and Use of Stellenbosch University (#1260) and permission to capture anurans, was granted under the permit of the Department of Environment and Forests, Andaman and Nicobar Islands (#CWLW/WL/134/350). The diet of anurans was determined using stomach flushing, a standard and low-risk technique to determine prey consumed (*Solé et al., 2005*). Anurans were hand-captured between 1800 to 2200 hrs; stomach flushing was carried out within 3 h of capture. We consciously avoided capture bias towards any particular size class, by actively searching for anurans of all size classes. As our sampling focussed on sub-adult and adult *H. tigerinus* and was completed in July (presumably before emergence of metamorphs) we did not examine the diet of metamorphs. In order to avoid mortality, we did not stomach flush individuals below 20 mm SVL and hence, individuals of co-occurring *Microhyla chakrapani* (ca. 10–30 mm SVL; *Pillai, 1977*) were not sampled. After excluding native anurans which did not co-occur with *H. tigerinus*, our samples included *Duttaphrynus melanostictus* (although its taxonomic and geographic status is uncertain, *Das, 1999*),

*Limnonectes* and *Fejervarya*. We conducted stomach flushing using a syringe (three ml to 10 ml for anurans of 20 mm–50 mm SVL and 60 ml for anurans >60 mm SVL), soft infusion tube, and water from the site of capture. In addition to SVL, we measured head width (HW) and lower jaw length (LJL) of the anurans, using a Vernier calliper (0.01 mm precision) and noted the sex. The stomach flushed individuals were toe-clipped (following *Grafe et al., 2011*) to record the total number of recaptures ($n = 54$). Individuals were released back to the capture site post completion of the procedure.

We collected the expelled prey items in a transparent beaker and sieved the contents using a mesh of 0.5 mm. Prey items from each individual were classified up to a minimum of order level, and further characterized by functional traits (hardness and motility, following *Vanhooydonck, Herrel & Van Damme, 2007*). Length and width of intact prey were measured under an 8x magnifying lens to the nearest 0.01 mm using a Vernier calliper and recorded along with the prey's life stage (adult/larvae). We preserved all prey items in 70% ethanol.

We also determined electivity of prey, based on prey consumption as compared to prey availability. Terrestrial prey were measured using five pitfall traps in each one ha plot, which were visited twice daily for a duration of three days (total of 30 trap occasions). Within each one ha plot, the pitfalls were arranged in the four corners and one in the centre of the plot. We used plastic traps, 80 mm in diameter and 300 mm high. A wet cloth was kept at the bottom to provide refuge to trapped animals, so as to prevent any predation before sample collection. We used chloroform soaked cotton balls to euthanize the invertebrate prey, prior to collection. These prey items were also identified up to the order level and measured for length and width. Our approach of estimating prey availability excludes flying evasive orders (e.g., adult lepidopterans) and vertebrate prey.

## Data analyses

We did not obtain adequate numbers of *Duttaphrynus melanostictus* ($n = 4$) individuals and hence they were not included in the analyses. We pooled samples from the three sites to examine diet at the species level for *H. tigerinus* and genus level for *Limnonectes* and *Fejervarya*. We assessed the number, volume, and frequency (number of individuals with a given prey item in their stomach) of consumed prey under each taxonomic category. Volume was calculated using the formula of an ellipsoid, following *Colli & Zamboni (1999)*,

$$\text{volume} = \frac{4}{3}\pi \left(\frac{l}{2}\right)\left(\frac{w}{2}\right)^2,$$

where, $l$ is prey length and w is prey width. Prey items for which volume could not be calculated due to lack of measurement data (i.e., fragmented prey) were assigned the median prey volume for that order. We carried out a generalized linear model to test the relationship between body size of anurans (SVL) and prey volume, after accounting for taxonomic identity of anurans. We log transformed SVL to adhere to the assumption of normality and cube root transformed prey volume, prior to the analysis.

In order to assess the overall importance of a prey category, based on the percentage of number, frequency and volume, we used the Index of Relative Importance (IRI, *Pinkas, Oliphant & Iverson, 1971*).

**Table 1 Sampling effort for diet assessment of the invasive *H. tigerinus* and native *Limnonectes* spp. and *Fejervarya* spp.** Sampling carried out in four habitat types across two seasons, at three sampling locations on the Andaman Islands.

| | Agriculture | | Plantation | | Disturbed forest | | Undisturbed forest | |
|---|---|---|---|---|---|---|---|---|
| | dry | wet | dry | wet | dry | wet | dry | wet |
| ***H. tigerinus*** | | | | | | | | |
| Karmatang | 41 | 35 | 29 | 29 | 0 | 0 | 0 | 0 |
| Webi | 32 | 35 | 48 | 38 | 0 | 0 | 0 | 0 |
| Wandoor | 0 | 0 | 38 | 33 | 0 | 0 | 0 | 0 |
| ***Limnonectes*** | | | | | | | | |
| Karmatang | 0 | 17 | 5 | 26 | 0 | 25 | 0 | 22 |
| Webi | 14 | 17 | 19 | 26 | 13 | 17 | 13 | 17 |
| Wandoor | 7 | 21 | 17 | 29 | 19 | 11 | 30 | 10 |
| ***Fejervarya*** | | | | | | | | |
| Karmatang | 0 | 0 | 0 | 0 | 0 | 0 | 0 | 0 |
| Webi | 1 | 0 | 0 | 0 | 1 | 0 | 0 | 0 |
| Wandoor | 19 | 17 | 13 | 2 | 10 | 0 | 2 | 0 |

To test for diet overlap, we employed the MacArthur and Levins' index *Ojk* (*MacArthur & Levins, 1967*) in the pgirmess package (*Giraudoux, 2016*); we built null models using the 'niche_null_model' function of the EcoSimR package (*Gotelli, Hart & Ellison, 2015*) to test for statistical significance of *Ojk*. We also assessed prey availability for each site across both dry and wet seasons, using the Simpson's diversity index (Supplemental Information 1). We determined electivity of terrestrial invertebrate prey by the anurans, using the Relativized Electivity Index (*Vanderploeg & Scavia, 1979*). Following *Measey (1998)*, we computed electivity for only those prey taxa with $n \geq 10$ prey items for *H. tigerinus* and *Limnonectes*; given the low sample size for *Fejervarya* (Table 1), we fixed the cut-off at $n \geq 5$. Further, electivity for *H. tigerinus* was calculated only for agriculture and plantations; electivity for *Fejervarya* was considered only for one site with adequate sample size: Wandoor (Table 1). All analyses were carried out in the statistical software R 3.4.1 (*R Core Team, 2017*).

## RESULTS

Overall, we sampled 798 individuals of the two native anurans and the invasive *H. tigerinus* (Table 1). We obtained 1,478 prey items (*H. tigerinus*: 687, *Limnonectes*: 618, *Fejervarya*: 173) belonging to 35 taxonomic categories in the stomach of 688 anurans (Table 2, Data S1). Vacuity index (i.e., proportion of empty stomachs) was higher in the dry season (19.68%) as compared to the wet season (8.67%). Less than 4% of prey items remained unidentified, mostly due to advanced levels of digestion. *H. tigerinus* consumed prey items under most of the taxonomic categories (29), followed by *Limnonectes* (25), and *Fejervarya* (14). Vertebrates were consumed by both *H. tigerinus* and *Limnonectes*, although the numeric and volumetric percentage of vertebrates consumed was higher for *H. tigerinus* (2.62%, 58.03%) than *Limnonectes*

**Table 2 Diet of *H. tigerinus* (n = 687), *Limnonectes* (n = 618) and *Fejervarya* (n = 173) in three sites on the Andaman archipelago.** Diet described in terms of percentage N, prey abundance; V, volume; F, frequency of occurrence in anurans, and IRI, Index of relative importance.

| Prey | *Hoplobatrachus tigerinus* (n = 687) | | | | *Limnonectes* (n = 618) | | | | *Fejervarya* (n = 173) | | | |
|------|------|------|------|------|------|------|------|------|------|------|------|------|
| | N% | V% | F% | IRI | N% | V% | F% | IRI | N% | V% | F% | IRI |
| Acari | 0 | 0 | 0 | 0 | 0.32 | 0.006 | 0.39 | 0.12 | 0.57 | 0.14 | 0.84 | 0.61 |
| Agamidae | 0.43 | 50.44 | 0.57 | 29.07 | 0 | 0 | 0 | 0 | 0 | 0 | 0 | 0 |
| Amphipoda | 0 | 0 | 0 | 0 | 0.48 | 0.07 | 0.58 | 0.32 | 0 | 0 | 0 | 0 |
| Anura | 0.87 | 4.95 | 1.14 | 6.65 | 0.32 | 5.12 | 0.39 | 2.12 | 0 | 0 | 0 | 0 |
| Aranae | 3.20 | 0.73 | 4 | 15.74 | 7.60 | 2.27 | 8.59 | 84.93 | 7.51 | 7.75 | 10.16 | 155.23 |
| Arthropoda | 6.55 | 0 | 8.57 | 56.22 | 5.50 | 0 | 6.64 | 36.53 | 0.57 | 0 | 0.84 | 0.48 |
| Blattaria | 1.45 | 0.33 | 1.90 | 3.42 | 1.29 | 0.71 | 1.56 | 3.14 | 0 | 0 | 0 | 0 |
| Chilopoda | 3.35 | 6.15 | 2.85 | 27.15 | 3.23 | 2.75 | 3.9 | 23.41 | 1.15 | 7.62 | 1.69 | 14.88 |
| Coleoptera | 29.73 | 12.14 | 24.57 | 1,029.14 | 15.85 | 10.34 | 15.42 | 404.29 | 9.24 | 20.50 | 12.71 | 378.16 |
| Brachyura | 0.58 | 2.40 | 0.76 | 2.27 | 0.16 | 0.81 | 0.19 | 0.19 | 0 | 0 | 0 | 0 |
| Dermaptera | 0.14 | 0.009 | 0.19 | 0.02 | 1.61 | 0.20 | 1.95 | 3.55 | 0 | 0 | 0 | 0 |
| Diplopoda | 0.87 | 0.07 | 0.76 | 0.72 | 3.55 | 0.73 | 3.12 | 13.41 | 0 | 0 | 0 | 0 |
| Diptera | 1.89 | 0.56 | 1.52 | 3.74 | 4.04 | 0.09 | 3.9 | 16.15 | 14.45 | 3.38 | 14.40 | 256.95 |
| Formicidae | 3.93 | 0.37 | 3.80 | 16.42 | 10.19 | 0.24 | 8.00 | 83.58 | 38.72 | 5.80 | 23.72 | 1,056.60 |
| Gastropoda | 4.22 | 0.71 | 4 | 19.76 | 3.23 | 1.5 | 3.32 | 15.72 | 0 | 0 | 0 | 0 |
| Geckonnidae | 0.14 | 0.45 | 0.19 | 0.11 | 0 | 0 | 0 | 0 | 0 | 0 | 0 | 0 |
| Hemiptera | 0.58 | 0.19 | 0.76 | 0.59 | 2.10 | 0.35 | 2.34 | 5.77 | 5.20 | 10.96 | 5.08 | 82.18 |
| Hymenoptera | 0.14 | 0.004 | 0.19 | 0.02 | 0 | 0 | 0 | 0 | 1.15 | 0.86 | 0.84 | 1.70 |
| Insecta | 1.45 | 0 | 1.90 | 2.77 | 1.29 | 0 | 1.36 | 1.76 | 6.35 | 0 | 9.32 | 59.27 |
| Isoptera | 2.62 | 0.24 | 2.09 | 6.01 | 7.44 | 1.88 | 4.49 | 41.89 | 2.31 | 0.87 | 3.38 | 10.81 |
| Lacertidae | 0.29 | 0.90 | 0.38 | 0.45 | 0 | 0 | 0 | 0 | 0 | 0 | 0 | 0 |
| Lepidoptera | 1.31 | 0.24 | 1.33 | 2.07 | 0.48 | 0.14 | 0.39 | 0.24 | 0 | 0 | 0 | 0 |
| Leplarva | 6.26 | 3.01 | 7.42 | 68.95 | 6.63 | 5.95 | 6.64 | 83.59 | 3.46 | 15.08 | 4.23 | 78.61 |
| Mantodea | 0.29 | 0.72 | 0.38 | 0.38 | 0 | 0 | 0 | 0 | 0 | 0 | 0 | 0 |
| Odonata | 0.72 | 0.07 | 0.95 | 0.76 | 0.16 | 0.04 | 0.19 | 0.04 | 0 | 0 | 0 | 0 |
| Oligochaeta | 1.31 | 0.77 | 1.52 | 3.18 | 4.69 | 54.54 | 4.10 | 242.95 | 0 | 0 | 0 | 0 |
| Opilionida | 0 | 0 | 0 | 0 | 0 | 0 | 0 | 0 | 0 | 0 | 0 | 0 |
| Orthoptera | 24.48 | 12.62 | 24.19 | 897.74 | 13.26 | 9.45 | 14.84 | 337.34 | 3.46 | 20.01 | 5.08 | 119.39 |
| Rodentia | 0.14 | 0 | 0.19 | 0.02 | 0 | 0 | 0 | 0 | 0 | 0 | 0 | 0 |
| Scincidae | 0.14 | 0.62 | 0.19 | 0.14 | 0 | 0 | 0 | 0 | 0 | 0 | 0 | 0 |
| Serpentes | 0.58 | 0.67 | 0.76 | 0.95 | 0.16 | 0.04 | 0.19 | 0.04 | 0 | 0 | 0 | 0 |
| Siphonaptera | 0 | 0 | 0 | 0 | 0 | 0 | 0 | 0 | 0.57 | 0.075 | 0.84 | 0.55 |
| Gastropoda | 0.29 | 0.27 | 0.38 | 0.21 | 0.80 | 1.97 | 0.78 | 2.17 | 0 | 0 | 0 | 0 |
| Unidentified | 1.89 | 0.26 | 2.47 | 5.35 | 5.33 | 0.69 | 6.44 | 38.87 | 5.20 | 6.92 | 6.77 | 82.19 |
| Zygentoma | 0 | 0 | 0 | 0 | 0.16 | 0.01 | 0.19 | 0.03 | 0 | 0 | 0 | 0 |

(0.48%, 5.16%; Table 2). Based on IRI, coleopterans and orthopterans constituted the major prey of *H. tigerinus* and *Limnonectes*, whereas, formicids and coleopterans formed the majority in the diet of *Fejervarya* (Table 2).

The diet of *H. tigerinus* overlapped significantly with that of *Limnonectes* ($Ojk = 0.87$, lower-tail $p > 0.999$, upper-tail $p < 0.001$) but there was no significant overlap with
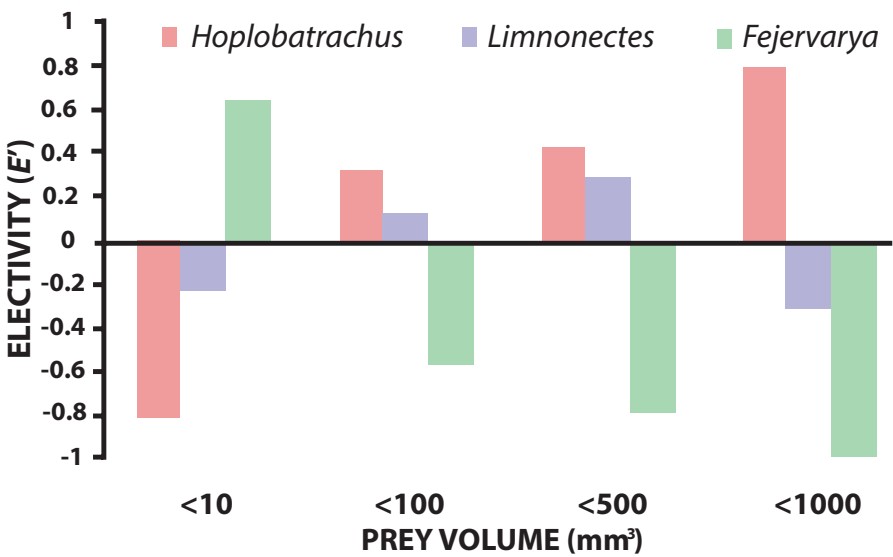

**Figure 3** **Prey electivity in terms of volume by the invasive *H. tigerinus* and native *Limnonectes* spp. and *Fejervarya* spp.** Prey electivity based on prey consumption and availability at three sites on the Andaman archipelago.

*Fejervarya* ($Ojk = 0.35$, lower-tail $p = 0.919$, upper-tail $p = 0.08$). The diet of the two native anurans overlapped significantly ($Ojk = 0.58$, lower-tail $p = 0.967$, upper-tail $p = 0.03$).

Based on availability of terrestrial invertebrates, prey electivity of all three anurans indicated a positive relationship between predator–prey body sizes (Fig. 3). While the largest species, *H. tigerinus*, strongly selected larger prey ($\geq 100$ mm$^3$), the smallest anuran, *Fejervarya*, selected for prey items smaller than 10 mm$^3$; the medium sized *Limnonectes* chose small and medium-sized prey items (10 mm$^3$–500 mm$^3$), although the magnitude of electivity (positive or negative) was lowest for this species (Fig. 1; Fig. 3). We found a positive correlation between prey volume and body size of *H. tigerinus* ($\beta = 1.93$, SE $= 0.21$, $p < 0.001$) and *Limnonectes* ($\beta = 0.88$, SE $= 0.25$, $p < 0.001$), but found no such relationship in case of *Fejervarya* ($\beta = -0.07$, SE $= 0.33$, $p = 0.83$). The majority of prey consumed by the three anurans was hard, and evasive, although diet of *Limnonectes* included a relatively higher proportion of soft and sedentary prey (Table 3). Terrestrial prey were the dominant type in the diet of *H. tigerinus* (91.29%), *Limnonectes* (93.18%), and *Fejervarya* (99.34%).

We observed several endemic vertebrate species in the diet of *H. tigerinus*, including the Andaman emerald gecko *Phelsuma andamanensis* ($n = 1$), Chakrapani's narrow mouthed frog *Microhyla chakrapani* (2), the Andaman skink *Eutropis andamanensis* (1), and Oates's blind snake *Typhlophs oatesii* (3). We also found *Limnonectes* (4), unidentified rodent (1), *Lycodon* sp. (1) and the invasive *Calotes versicolor* (3) in the diet of *H. tigerinus* (Data S1). *Limnonectes* preyed upon a conspecific on one occasion and an unidentified anuran in another instance.

**Table 3 Prey electivity ($E'$) of the invasive _H. tigerinus_ and native _Limnonectes_ and _Fejervarya_ based on prey hardness and motility in three sites of the Andaman archipelago.** Classification of prey hardness and motility following _Vanhooydonck, Herrel & Van Damme (2007)_. Sampling carried out in four habitat types across two seasons, at three sampling locations on the Andaman Islands.

| _H. tigerinus_ | | | | | |
|---|---|---|---|---|---|
| | dry | wet | | dry | wet |
| soft | −0.10 | −0.31 | sedentary | −0.12 | −0.22 |
| medium | 0.80 | −0.07 | medium | −0.70 | 0.20 |
| hard | −0.59 | 0.32 | evasive | 0.85 | −0.01 |
| _Limnonectes_ | | | | | |
| | dry | wet | | dry | wet |
| soft | 0.52 | 0.14 | sedentary | 0.41 | 0.15 |
| medium | 0.15 | −0.09 | medium | −0.46 | −0.11 |
| hard | −0.52 | −0.09 | evasive | 0.31 | −0.06 |
| _Fejervarya_ | | | | | |
| | dry | wet | | dry | wet |
| soft | 0.14 | −0.18 | sedentary | 0.01 | −0.33 |
| medium | −0.45 | −0.43 | medium | 0.10 | 0.49 |
| hard | −0.01 | 0.38 | evasive | −0.34 | −0.45 |

## DISCUSSION

We expected the diet of invasive _H. tigerinus_ to overlap significantly with the diet of the two native anurans considered. However, we found a significant overlap only with _Limnonectes_, such that when prey is limited competition may arise. As expected, small vertebrates constituted a majority of _H. tigerinus_ diet by volume, suggesting potential impact by predation on a large proportion of the endemic island fauna. Volume of prey elected was positively related to predator size (Fig. 3); within species, volume of prey consumed was positively correlated with predator size for _H. tigerinus_ and _Limnonectes_ only.

We observed 86% niche overlap between _H. tigerinus_ and _Limnonectes_, which was statistically significant in comparison to the constructed null model; whereas, niche overlap of _H. tigerinus_ with _Fejervarya_ was not significant. On the other hand, prey electivity suggests that _H. tigerinus_ strongly elected for medium-sized and larger prey whereas small and medium-sized prey were elected by _Limnonectes_ (Fig. 3). This may result in competition for prey ranging from 10–500 mm$^3$ between the two anurans, under the conditions of limited prey. Trophic competition in amphibians may lead to a decrease in fitness (e.g., growth rate) and affect population level processes (_Benard & Maher, 2011_). The impact of invasive amphibians (post-metamorphic) via trophic competition has been documented in fewer studies as compared to predation (_Measey et al., 2016_), but this mechanism may affect taxa at various trophic levels (_Smith, Beard & Shiels, 2017_). Metamorphs of _H. tigerinus_ may also compete with both _Fejervarya_ and _Limnonectes_ as they would fall under the same size class (20 mm–40 mm; _Daniels, 2005_). The observed positive correlation between body size and prey volume in the case of both _H. tigerinus_ and _Limnonectes_ also supports the notion that metamorphs of these species may compete for

small prey. Although our sampling did not evaluate the diet of *H. tigerinus* metamorphs, we think this may be relevant as competition between juvenile *Lithobates catesbeianus* and small native anurans has been previously documented on Daishan Island, China (*Wu et al., 2005*).

Evaluating dietary overlap is a pre-cursor to determining trophic competition due to invasive populations, which do not have shared evolutionary history with native species. Dietary overlap in co-occurring species may be independently influenced by prey availability (*Kuzmin, 1995*), prey taxa (*Lima, 1998*), prey size (*Toft, 1981*; *Vignoli, Luiselli & Bologna, 2009*; *Crnobrnja-Isailović et al., 2012*) and a combination of these factors. Therefore, it is essential to design studies and interpret dietary patterns with reference to all three factors, in order to arrive at meaningful inferences on prey consumed, dietary overlap, and probable subsequent competition (Kuzmin, 1990; but see *Kuzmin, 1995* regarding criteria for competition). Further, prey size should ideally be measured in terms of volume, as it is known to be a better dietary descriptor (*Vignoli, Luiselli, 2012*).

*H. tigerinus* preyed upon three classes of vertebrates (Amphibia, Reptilia, and Mammalia), which accounted for a significant proportion of its diet by volume, although vertebrate prey was numerically inferior to invertebrates in the diet. Such major contribution to the volume of prey by vertebrates (despite numerical inferiority) has been observed for *Lithobates catesbeianus* and *Xenopus laevis* (*Boelter et al., 2012*; *Vogt et al., 2017*); anurophagy may also contribute significantly to the diet of many amphibians (*Measey et al., 2015*; *Courant et al., 2017*). We observed several endemic species in the diet of *H. tigerinus*, which may become threatened if frequently preyed upon. *Limnonectes* was also consumed by *H. tigerinus*, thereby indicating a potential two-pronged impact through predation and competition. However, demographic change (if any) in *Limnonectes*, due to predation and competition by *H. tigerinus*, was not evaluated in this study. The invasive *H. tigerinus* on the Andaman Islands reportedly consume poultry (M Chandi, 2017, pers. comm.; *Mohanty & Measey, in press*) and stream fish (NP Mohanty, 2017, unpublished data), resulting in a potential economic impact. We expect the invasive *H. tigerinus* on Madagascar (*Glaw & Vences, 2007*) to similarly consume a large proportion of vertebrates in its diet and consider the invasion to be a threat to the highly diverse small vertebrates of Madagascar.

Despite the presence of a large portion of vertebrates in the diet of *H. tigerinus*, its trophic position (consistency of vertebrate prey consumption) can only be ascertained with stable isotope analyses (*Huckembeck et al., 2014*). Although, diet analysis of invasive species can identify vulnerable taxa and confirm at least 'minimal' to 'minor' levels of impact through predation and competition (*sensu Blackburn et al., 2014*; *Hawkins et al., 2015*), such analysis must be complimented with evidence of trophic level effects to evaluate the degree of impact (*Smith, Beard & Shiels, 2017*).

The large proportion of ants in the diet of *Fejervarya* does not necessarily prove specialization for myrmecophagy. *Hirai & Matsui (2001)* inferred relatively weaker avoidance of ants by *Glandirana rugosa* as compared to other anurans. Although we found the same pattern for *Fejervarya* based on prey electivity ($E' = -0.02$), it does not prove weak avoidance either. As social insects, ants may be disproportionately captured in the pitfall

traps; therefore, it is necessary to compliment diet studies on potentially myrmephagous predators with additional evidence (e.g., cafeteria experiments). *H. tigerinus* and *Fejervarya* chose evasive prey, suggesting that these two species are mostly ambush ('sit and wait') predators; *Limnonectes* elected sedentary prey along with other prey types, indicating a combination of 'active search' and 'sit and wait' foraging (Table 3; *Huey & Pianka, 1981*; *Vanhooydonck, Herrel & Van Damme, 2007*). Generally, soft bodied prey are considered to provide more nutrition by size as compared to hard prey and, therefore, it is hypothesized that species will select soft prey more often than hard prey, which in turn is dependent on prey availability by season (*Measey et al., 2011*; *Carne & Measey, 2013*). However, we find that diet does not appear to vary considerably across the seasons and is governed more by size than hardness of prey (Fig. 3; *Werner, Wellborn & McPeek, 1995*).

Although our sampling for diet analysis by stomach flushing was adequate (Table 1), our assessment of prey availability did not include flying invertebrates and vertebrates, which prevents us from carrying out electivity analyses on these taxa.

## CONCLUSION

Diet analyses of *H. tigerinus* confirmed our first hypothesis, i.e., significant predation of *H. tigerinus* on endemic vertebrates (hypothesis 1) and partially supported the second hypothesis of a high diet overlap with native anurans (hypothesis 2) indicating potential competition; overlap was significant only for the large-bodied *Limnonectes*. Given the observed high density of *H. tigerinus* in human modified habitats on the Andaman archipelago (NP Mohanty, 2017, unpublished data), trophic competition and predation by *H. tigerinus* may have a significant impact on native anuran populations in these habitats. Pursuing our additional aim of characterizing anuran foraging modes, we determined the foraging strategy of *H. tigerinus* and *Fejervarya* as ambush foraging ('sit and wait') and that of *Limnonectes* to be a combination of 'active search' and 'sit and wait' foraging. In addition to quantifying the trophic niche of anurans belonging to three genera, we stress the necessity to evaluate prey availability and volume in future studies for meaningful insights into diet of amphibians.

## ACKNOWLEDGEMENTS

We would like to thank the Department of Environment and Forests, Andaman and Nicobar Islands for granting permits, the Andaman & Nicobar Environment Team (ANET) for facilitating field work, Saw Issac and Saw Sathaw for collecting part of the data and help during field work. NPM would like to acknowledge the support and advice of Dr. Karthikeyan Vasudevan, Dr. Harikrishnan S., Dr. Manish Chandi, and Ashwini Mohan during the study.

### Funding

This study was supported by the DST-NRF Centre of Excellence for Invasion Biology (CIB). The Inlaks Shivdasani Foundation-Ravi Sankaran Fellowship Programme, the Rufford Small Grants (#20818-2), and the Department of Botany and Zoology, Stellenbosch University all contributed funding. The funders had no role in study design, data collection and analysis, decision to publish, or preparation of the manuscript.

### Grant Disclosures

The following grant information was disclosed by the authors:
DST-NRF Centre of Excellence for Invasion Biology (CIB).
Inlaks Shivdasani Foundation-Ravi Sankaran Fellowship Programme.
Rufford Small Grants: #20818-2.
Department of Botany and Zoology, Stellenbosch University.

### Competing Interests

John Measey is an Academic Editor for PeerJ.

### Author Contributions

- Nitya Prakash Mohanty conceived and designed the experiments, performed the experiments, analyzed the data, contributed reagents/materials/analysis tools, prepared figures and/or tables, authored or reviewed drafts of the paper, approved the final draft.
- John Measey conceived and designed the experiments, contributed reagents/materials/-analysis tools, authored or reviewed drafts of the paper, approved the final draft.

### Animal Ethics

The following information was supplied relating to ethical approvals (i.e., approving body and any reference numbers):

The research protocol was approved by the Research Ethics Committee: Animal Care and Use, Stellenbosch University (#1260).

### Field Study Permissions

The following information was supplied relating to field study approvals (i.e., approving body and any reference numbers):

Permission to capture anurans was granted under the permit of the Department of Environment and Forests, Andaman and Nicobar Islands (#CWLW/WL/134/350).

### Data Availability

All dietary data produced in this project are available in the Supplemental File.

### Supplemental Information

Supplemental information for this article can be found online at http://dx.doi.org/10.7717/peerj.5698#supplemental-information.

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
