# Peer review of "What’s for dinner? Diet and potential trophic impact of an invasive anuran Hoplobatrachus tigerinus on the Andaman archipelago"

_PeerJ, doi:10.7717/peerj.5698_

## Round 0.1 · original submission · Minor Revisions

The reviewers and I all agree that, with some revisions, your manuscript will pass the standards established in PeerJ's editorial criteria. The reviewers both had a number of suggestions that I would like to see addressed in the revision. Please pay particular attention to Reviewer 1's comments on: tempering the manuscript by including "potential" in the title; suggestion to remove the niche breadth analysis for the reasons we discussed prior to review (lack of ability to make statistical inference at this time); suggestion of examining relationship between prey and predator size; and numerous typographical and reference related suggestions in the attached annotated manuscript. Please also pay particular attention to Reviewer 2's comments on further formalizing knowledge of this invader in particular and anuran invaders in general in the introduction, including reference to other potential invader impacts (e.g., serving as sources or reservoirs for chytrid fungus). Finally, Reviewer 2 pointed out that the manuscript mentions the limitation of the method of assessing prey availability doesn't capture vertebrates, but vertebrates appear to make up a large portion of the diet of H. tigerinus.

·

Basic reporting

This interesting study surely deserves publication, since it deals with an anuran species that is not ranked among the widely recognized invasive amphibians, and hence it adds important information to the current knowledge about amphibian invasions. Moreover, the diets of two native anurans were summarized for the first time in the Andaman Islands, adding important information about the biology of these species. I suggest the inclusion of the word “potential” in the title (“…Diet and potential trophic impact…”) since no direct measure of negative impact (e.g. demographic change, survival rate) has been performed. Some statements by the authors in the discussion also concur with this interpretation (e.g. lines 288-289, 323). The language is good and professional, but some improvements could be made on some sentences (e.g., lines 85-90, 304-305, 320-321). The literature cited is sufficient, but I felt that a stronger background is required to support the second hypothesis investigated by the authors (see comments in the reviewed manuscript I have uploaded). The article structure conforms to the “standard sections”, although headings were not provided for the abstract. Figures are relevant, but I made some suggestions to improve their appearance and readability (see comments in the reviewed manuscript). The study contains results that are conclusive in relation to the two hypothesis stated; however, in my opinion the niche breadth analysis added no relevant information, since the temporal sampling was too limited to perform statistic comparisons, and the electivity and prey size investigations were sufficient in providing basis for the interpretation of niche overlap. Additionally, the authors poorly mentioned niche breadth in the discussion, so that I suggest this analysis to be removed from the study with no detrimental outcomes. Alternatively, in the case of maintaining this analysis a hypothesis about niche breadth should be stated, and a deeper discussion about niche breadth results ought to be included.

Experimental design

This study fits in the “Aims and Scope” of PeerJ, in both Biological and Environmental Sciences. The knowledge gap was clearly identified, and the research questions were well defined in the last paragraph of the introduction with sufficient background. However, the same is not true for the abstract, where a clearer statement of the research questions should be made. The authors provided an ethical approval statement, as well as the number of the capture permit (lines 135-138). The Methods were described in sufficient detail for most of the analyses; although some information should be added to clarify how recaptured frogs were considered during the estimation of prey frequencies (see my comment at line 154). Moreover, a correlative analysis designed to assess how prey size (volume) is related to frog size and to frog species should be included to better support part of the conclusions (see my comment at line 248).

Validity of the findings

I thank the authors for providing the raw dataset in a comprehensive format. Data are robust and statistically sound. Speculation was made, but well stated as such, on lines 263-268. The conclusions, however, need some improvements to be fully connected to the original questions, hypothesis and goals of the study (see my comments on lines 318-323). For example, no conclusive sentence has been done about predation behaviour, even though describing this behaviour was part of the aims of the study. Again, I ask the authors to perform a statistical analysis confronting prey size (volume) and frog size, accounting for interspecific effects (i.e., differences related to frog species), to better support the relationship between prey size and trophic niche overlap.

Additional comments

Further comments, suggestions and demands that I made can be found in the reviewed manuscript that I have added to this review. Pay special attention to the list of references; I found some disagreements between listed works and citations in the text (e.g., different years). I also recomend a revision in the literature cited to assure that all reference entries conform to the journal stile.

Reviewer 2 ·

Basic reporting

The whole manuscript is generally well written. However, the authors used several unpublished references throughout the whole text. Please check that whether it is allowed by the journal. In addition, the authors need provide more background knowledge on amphibian impacts and the specific study species here. Please see more detailed comment below in "General comments for the author" section.

Experimental design

The field sampling and survey is well done. However, the authors need more clarification on how they estimated the residence time of this frog invader in the study area. Please see my detailed comment below in the "General comments for the author" section.

Validity of the findings

Data and findings are robust. However, the authors need more clarifications on why they excluded the vertebrate preys from the diet preference analyses. Please see my detailed comment below in the "General comments for the author" section.

Additional comments

This work is very interesting since the authors focused on the diet composition and prey selection of the invasive Indian bullfrog in the Andaman archipelago, which is one global biodiversity hotspot. Compared with many researches investigating diet of the other invasive amphibians such as the American bullfrog across the globe, we still understand so little about the predation impact of the Indian bullfrog. Here, the authors found that this invasive frog in the Andaman archipelago can predate on native endemic vertebrates, and there is a high diet overlap of this invader with native Limnonectes spp., indicating its potential diet competition with native frogs. The authors further identified some preferred invertebrate preys of the Indian bullfrog after accounting for the prey availability in environment. The results of the present study are somewhat novel for this species and have important implications for conservation of native species in the Andaman Islands. The whole manuscript is generally well written. Following are my concerns, which may be helpful to the improvement of the manuscript.

Line 38-46, the descriptions on impacts of invasive amphibians need more details to increase readability of this work. For example, except for predation and competition, which are two most direct impacts on native species, invasive amphibians can also have evolution impacts on native frogs through reproductive interference (e.g. D’Amore et al. 2009, Theis & Caldart 2015) and transmission of amphibian diseases such as the notorious Chytrid fungus Bd (e.g., Garner et al. 2006, Liu et al. 2013).

Line 44, for documented predation impact on amphibians, please see also Liu et al. 2015, which provides direct evidence on the preference of the invasive American bullfrog on Chinese endemic frogs after accounting for the availability of different prey species in the environment.

Line 62-63, how about the impacts of Indian bullfrog on native amphibians in Madagascar, Maldives and Laccadive Islands? Are there more background knowledge on this invasive species?


Line 125: the authors need provide more detailed information on how they estimated the residence time (e.g., at least 3 years here) of the bullfrogs established in the two study sites. In addition, the authors need to give more information on the introduction or invasion history of this species in the study area.

Line 168: the authors need more explanations on why you excluded the vertebrate prey species from the diet selectivity analyses. In my opinion, the predation of bullfrogs on native vertebrates such as native frogs are very important for the evaluation of bullfrog impacts on native biodiversity especially the authors found that small vertebrates consisted a majority of the bullfrog diet in the study site. In addition, vertebrate preys can greatly influence the predation choice of the invasive bullfrogs on native species because vertebrates generally have higher biomass than invertebrates.

Suggested references

D’Amore et al. Reproductive interference by an invasive species: an evolutionary trap. Herpetological Conservation and Biology 4.3 (2009): 325-330.
Garner et al. The emerging amphibian pathogen Batrachochytrium dendrobatidis globally infects introduced populations of the North American bullfrog, Rana catesbeiana. Biology letters 2.3 (2006): 455-459.
Liu et al. Climate, vegetation, introduced hosts and trade shape a global wildlife pandemic. Proc. R. Soc. B 280.1753 (2013): 20122506.
Liu et al. Diet and prey selection of the Invasive American bullfrog (Lithobates catesbeianus) in southwestern China. Asian herpetological research 6 (2015): 34-44.
Theis & Caldart. Multiple interspecific amplexus between a male of the invasive Bullfrog Lithobates catesbeianus (Ranidae) and two males of the Cururu toad Rhinella icterica (Bufonidae). Herpetology Notes8 (2015): 449-451.

---

## Round 0.2 · Minor Revisions

Thanks for your hard work on revising the manuscript. The reviewers have some truly minor revisions that I would like to suggest. Please pay particular attention to Reviewer 2's request to completely remove Niche Breadth from the Methods and Results, and consider adding text regarding recapture events to the results. Following your revision, I do not anticipate needing to send it out again and we should be able to turn this around fairly quickly.

Many thanks,
Ben

·

Basic reporting

My original positive comments about the manuscript are still valid here. The manuscript was much improved over the original draft, which indicates the authors worked thoroughly to accommodate the reviewers’ suggestions and constructive criticisms. Almost all of my suggestions have been addressed and my questions answered. However, the position of the authors about the niche breadth analysis remains unclear to me. They removed the original figure about niche breadth and mentioned in the response letter that they had removed this analysis from the text. Nevertheless, I found mention to niche breadth in the Material and Methods (lines 201-206) and Results (line 233), where they just stated “Niche breadth (J’) was highest for Limnonectes, followed by H. tigerinus, and Fejervarya”, with no quantitative data. In contrast, niche breadth was removed from the Abstract and Discussion. I maintain my suggestion to remove niche breadth from the text, particularly now when the authors decreased the emphasis given to it in this revised version of the manuscript. Please see additional comments below in "General comments for the author" section.

Experimental design

In the response letter, the authors answered my question about how they had dealt with recaptured frogs; they had 54 recaptures and considered recaptured individuals as independent samples (this is my interpretation of their response; my apologies if it is not correct). I think they should add this information to Material and Methods (lines 166-168), or place it in the Results.

Validity of the findings

The conclusions of the study are now much more informative and aligned with the hypothesis and research purposes, reinforcing the validity of the findings.

Additional comments

-line 26: remove “the genus” before Fejervarya; in my opinion it is not necessary;
-line 165: word “length” is repeated;
-line 241: remove the signal “<” before 10 mm3;
-line 255: I expected to see more details about the two predation events of anurans by Limnonectes informed in this line; for example, the identities of the preyed species and the dates of recording. I think this sentence is of poor value in the current format, so please provide more information.

Reviewer 2 ·

Basic reporting

no comment

Experimental design

no comment

Validity of the findings

no comment

Additional comments

The authors have addressed all the comments from my first round of review. As previously stated, I think this manuscript covers an interesting and important topic in amphibian invasion ecology, that deserves publication in this Journal. I recommend this article for publication.

---

## Round 0.3 · accepted · Accept

My sincerest thanks for your revisions and email, and apologies for missing that they had come in.

#